# Endogenous Neurostimulation and Physiotherapy in Cluster Headache: A Clinical Case

**DOI:** 10.3390/brainsci9030060

**Published:** 2019-03-12

**Authors:** Gonzalo Navarro-Fernández, Lucía de-la-Puente-Ranea, Marisa Gandía-González, Alfonso Gil-Martínez

**Affiliations:** 1Departamento de Fisioterapia, Centro Superior de Estudios Universitarios La Salle, Universidad Autónoma de Madrid, Madrid 28023, Spain; gonza_navarro93@hotmail.com (G.N.-F.); luciadelapuenteranea@gmail.com (L.d.-l.-P.-R.); 2Motion in Brains Research Group, Instituto de Neurociencias y Ciencias del Movimiento, Centro Superior de Estudios Universitarios La Salle, Universidad Autónoma de Madrid, Madrid 28023, Spain; 3Servicio de Neurocirugía, Hospital Universitario La Paz, Madrid 28046, Spain; marisagg4@hotmail.com; 4Hospital La Paz Institute for Health Research, Madrid 28046, Spain

**Keywords:** cluster headache, neurosurgery, pressure pain threshold, physiotherapy

## Abstract

Objective: The aim of this paper is to describe the progressive changes of chronic cluster headaches (CHs) in a patient who is being treated by a multimodal approach, using pharmacology, neurostimulation and physiotherapy. Subject: A male patient, 42 years of age was diagnosed with left-sided refractory chronic CH by a neurologist in November 2009. In June 2014, the patient underwent a surgical intervention in which a bilateral occipital nerve neurostimulator was implanted as a treatment for headache. Methods: Case report. Results: Primary findings included a decreased frequency of CH which lasted up to 2 months and sometimes even without pain. Besides this, there were decreased levels of anxiety, helplessness (PCS subscale) and a decreased impact of headache (HIT-6 scale). Bilateral pressure pain thresholds (PPTs) were improved along with an increase in strength and motor control of the neck muscles. These improvements were present at the conclusion of the treatment and maintained up to 4 months after the treatment. Conclusions: A multimodal approach, including pharmacology, neurostimulation and physiotherapy may be beneficial for patients with chronic CHs. Further studies such as case series and clinical trials are needed to confirm these results.

## 1. Introduction

The cluster headache (CH) has been defined by the International Headache Society as the most frequent trigeminal autonomic headache [1]. CHs is reported to be found in 0.12% of the population, with an overall male-to-female ratio of 3:1; episodic CH is more common than chronic CH [2,3]. CHs are characterized by severe unilateral pain in short-duration episodes, which are associated with ipsilateral autonomic symptoms which primarily encompass the temporal, supraorbital and infraorbital areas [1,3]. 

Much research performed in the early 2000s confirmed the involvement of the changes in the hypothalamus in patients with CH. For instance, May et al. concluded that there were activation and structural changes in the gray matter of the posterior and inferior parts of the hypothalamus [4]. More recent studies have also shown that there is hypothalamus activation in patients with CH, but there are no structural changes in the gray matter in that area, unlike other regions in which morphological changes were found, such as the hippocampus, anterior insula, orbitofrontal cortex, cerebellum, temporal lobe, anterior cingulate cortex and the primary and secondary somatosensory cortex [5]. The convergence theory describes the anatomo-functional convergence of the cervical (especially C2), somatic trigeminal and dural trigeminovascular afferent neurons on second-order nociceptors in the trigeminocervical complex [6]. One of the supporting studies found limited duration, frequency and intensity of cluster attacks after blockades, asserting the theory of convergence [7,8,9].

These findings could, therefore, be related to the extensive effect of pharmacological treatments proposed for patients with CH, such as sumatriptan, [3,10,11] other triptans, [12,13] verapamil, [13] lithium, [3,13] oxygen therapy, [14] onabotulinumtoxinA [15] and alternative treatments such as valproic acid, topiramate and methylprednisone [13]. 

The American Headache Society only proposes pharmacological and oxygen therapy as acute treatment with “A” level recommendation [16]. The lack of evidence makes the inclusion of other therapies in the evidence-based treatment guidelines more difficult. Moreover, the review of the effects of the principal drugs used in the treatment of acute CH shows side effects such as chest pain, paresthesia, dizziness, tingling feeling or numbness of the limbs, heaviness, asthenia, nausea, unpleasant taste, and somnolence [17,18]. Somatostatin treatment also has some side effects, and the most frequent among them are hyperglycaemia, nausea, abdomianal pain, diarrhea, and meteorism [18]. Adverse effects are also present in prophylactic treatments. Lithium carbonate produces tremor, gastrointestinal disturbances, dizziness, olfactory disorders and polyuria in CH patients [18,19,20]. Even though verapamil has been considered as the first-line prophylactic drug for the treatment of CH patients with “A” level recommendation [13], it should be used carefully, because verapramil treatment is noted to have a correlation with the electrocardiographic abnormalities (19% arrhythmias and 36% of bradycardia incidence) [21]. 

Limitations of drug treatment are advised in both episodic and chronic CH because 10% to 20% of CCHs have the chance of becoming drug-resistant headaches [22], and a new line in the research has been opened to increase the use of non-pharmacological adjuvant treatments for CH.

There are some non-invasive stimulation methods that have been used in CH patients. At first, the results obtained by Nesbitt et al. in 2015 suggested that non-invasive vagal nerve stimulation can be used as both acute and prophylactic treatment in ECH and CCH [23]. However, it has recently been shown that non-invasive vagal nerve stimulation is an effective treatment in ECH when compared with sham stimulation, but not in the CCH patient group [24]. Additionally, some investigators are assessing the effects obtained by using transcranial magnetic stimulation techniques in CH [25,26], but the evidence is still limited [27]. 

One of the most investigated invasive stimulation methods is sphenopalatine ganglion stimulation (SPG). This method was used in a study published in 2017, and it was concluded that after 24 months, 45% of the patients with refractory CCH were acute responders and 35% were frequent responders. Even though 81% of the side effect incidences were reported, they noted that SPG produced 61% of therapeutic responses and should be considered as an interesting treatment for refractory CCH [28]. Another invasive stimulation method commonly used in CH patients is occipital nerve stimulation (ONS).Although it has shown that ONS can reduce the frequency and intensity of headache at least by a 50% in about 60% of the patients [29], there is a lack of randomized controlled trials of ONS for the treatment of CH [27]. 

Moreover, manual therapies are being investigated in other primary headache cases; studies have shown that manual therapy was significantly better in the control group in reducing the intensity and frequency of headache [30]. 

Our aim was to describe the management of chronic CH in a patient who is treated by a multimodal approach including pharmacology, neurostimulation and physiotherapy.

## 2. Methods

The CARE (case reporting guideline development) checklist was used to prepare this case report and to complete it to improve quality reports in clinical cases [31].

### 2.1. Patient

A male patient of 42 years of age was diagnosed with left-sided refractory chronic CH by a neurologist in November 2009. The following pharmacological treatment was prescribed by the neurologist: (1) preventive treatment: verapamil (80 mg), topiramate (100 mg), escitalopram (15 mg), clonazepam (0.5 mg), lithium (100 mg) and prednisone (25 mg); and (2) abortive treatment: subcutaneous sumatriptan injection (6 mg) and 100% oxygen. The patient reported loss of memory and concentration as an adverse effect of the medication. In June 2014, the patient underwent a surgical intervention in which a bilateral occipital nerve neurostimulator was implanted as the treatment method for headache. 

### 2.2. Neurostimulator Implantation Procedure

To establish a subcutaneous stimulation of the greater occipital nerve complex, the implantation of bilateral 8-contact occipital lead was performed, using anatomical and radiological landmarks. 

Surgery was performed with the patient lying in the prone position and completely awake. Under local anesthesia, a midline incision in the back of the neck was made with a small epifascial scalpel. Guided by an anteroposterior X-ray image, a subcutaneous needle followed the curvature of the skin from the C1–C2 transition to the mastoid process on one side, avoiding the perforation of the fascia to prevent subfascial electrode localization on the other side.

The electrodes were placed subcutaneously and the needles were removed. The electrodes were anchored in the fascia after checking the correct coverage of the target area, and then two loops were placed. The electrodes were tunneled to the subcutaneous point at the halfway point between the midline neck incision and the subcutaneous gluteal buttock (on the right or left side, depending on the patient’s preference), where the another set of loops lay, which were attached to each electrode and then connected with the extensions (Figure 1). To identified the greater occipital nerve complex correctly, some serial ultrasound images and videos published by Chang KV et al. were used [32].

The extensions were tunneled to the subcutaneous gluteal buttock, where the implantable pulse generator was definitively placed after the extensions were connected and the correct impedances and stimulation were verified. Incision closure was performed when the correct hemostasis had occurred.

### 2.3. Evaluation

After signing the informed consent, the patient was evaluated before the treatment procedure, after the treatment, and at the 3rd month and 4th month after the treatment. A blinded experienced evaluator assessed both physical and psychological variables, which were selected according to previous observational studies in patients with CH [33]. Physical variables were pressure pain thresholds (PPTs) assessed bilaterally in the cranial and extracranial areas with a digital algometer [34] (Fx. 25 Force Gage, Wagner Instruments, Greenwich, CT, USA); two-point discrimination was tested in the trigeminal areas with an esthesiometer [35]; and cervical flexor endurance was measured with a craniocervical flexion test [36]. Psychological variables were as follows: impact of headache on the quality of life and work performance (HIT-6 scale) [37], pain catastrophizing (PC Scale) [38], neck disability (neck disability Index) [39] and depression symptoms (Hamilton depression rating scale) [40]. At each assessment, we measured the impact and outcome of disease in the patient’s daily life (with a verbal numeric rating scale) [41], the intensity and frequency of pain, crisis duration and medication used in each episode, and all the data were recorded in a diary. 

After observing the values obtained in the first assessment session, such as the reduced endurance of deep flexor cervical muscles (Table 1), the authors decided to include physical therapy as an adjuvant treatment for the patients.

### 2.4. Postsurgical Physiotherapy Approach

The patient received 8 physical therapy sessions in 6 weeks. Manual therapy techniques consisted of passive mobilization of the anterior–posterior upper cervical region, which can directly influence the upper three cervical segments (C0–C3), and three sets of 2 minutes of mobilization were performed with 0.5 Hz frequency and 30 seconds of rest [42]; additionally, the neurodynamic mobilization of the trigeminal nerve was performed, with 30 repetitions with 0.5Hz frequency (10 global mobilizations of mandibular opening and 10 mobilizations on each side of the mandibular laterotrusion to ease the tension to the auriculotemporal nerve) (Figure 2A,B). Finally, home exercises were prescribed to improve the motor control of deep flexor muscles, as described by Harris et al., [43], and the patient was advised to perform three sets of 10–20 repetitions including 5–10 seconds of contraction with 5–10 seconds of relaxation, depending on the patient´s strength. 

## 3. Results and Discussion

This is the first case to be reported to receive treatment as a combination of neurostimulation and physiotherapy approaches for chronic CH. After analyzing the patient´s case diary of headache, a decreased frequency of CH attacks was observed at the end of treatment, and the progress was maintained even 4 months after the treatment; additionally, it was reported that for the first 2 months there was no pain at all (Table 2). These results could support the use of a multimodal approach as an adjunctive treatment for occipital nerve neurostimulation (C2–C3). The benefits obtained with neurostimulation treatment might support the trigeminal–cervical convergence hypothesis [44,45], according to which craniofacial pain is generated by the activation of second-order neurons from the trigeminal nucleus caudalis, [45] a mechanism generated both in animals and in humans. [46] Other authors have developed studies in which it was observed that stimulation of the superior sagittal sinus, innervated by the ophthalmic branch of the trigeminus, also take part in the stimulation of second-order neurons of the trigeminal nucleus caudalis and the dorsal horn of C2–C3. Thus, this supports the trigeminal–cervical convergence hypothesis and provides the basis for neurostimulation. 

In addition, physical therapy techniques such as manual therapy, therapeutic exercise or a combination of both, specifically applied in the upper cervical region, had already caused a decrease in the frequency and intensity of various primary headaches. [47,48,49] These results reinforce our hypothesis that the combination of these techniques might reduce headache frequency due to the activation or inhibition of sensory input from the trigeminal–cervical system [46] in chronic CH. 

The decrease in frequency of attacks after the multimodal program can lead to decreased levels of anxiety, helplessness (PCS subscale) and impact of headache (Table 1). Ruscheweyh et al. demonstrated that the frequency of a primary headache is associated with pain-specific disability, quality of life, anxiety and depression, which are significantly more pronounced in patients with a chronic headache than in patients who have episodic attacks [50].

Moreover, the patient showed an improvement (increase) in most PPTs bilaterally, except in those that were examined and assessed on the greater occipital nerve (Table 1). These results could be due to the analgesic effect generated by the joint mobilization [42,51] or trigeminal neural mobilization, [52] along with the pharmacological and neurostimulation treatments. However, the decrease in the occipital nerve PPT coincides with the area of neurostimulator implantation, where the patient had reported a feeling of numbness (paresthesia) [44] that increased while performing a passive lateral glide of C0–C1, which suggests that active and passive craniocervical flexion could generate allodynia at these points, as occurs in CH conditions [53].

Finally, the inclusion of a therapeutic exercise program could also cause an increase in the strength and motor control of the cervical muscle, which was proved by comparing the pretreatment assessment and the assessment at the fourth month after the treatment (Table 1), which is supported by the evidence of the fact that stabilization exercise is effective in reducing pain [54] and might be useful in the treatment of other primary headaches as well [55]. Instead of longer follow-up, it is better to make the patient aware of the fact that the combination of neurostimulation and physiotherapy could be effective in reducing the frequency, duration, and severity of the CH. This may remove their lack of adherence to the treatment.

Finally, the limitation should be marked that, currently, ultrasound is widely used in assessing neuromuscular disorders. Some headaches could result from the chronic affection of the para-spinal and sub-occipital muscles, which can be easily checked by using ultrasound [32]. Unfortunately, the authors did not use this to examine possible musculoskeletal painful origins in this single case.

In conclusion, a multimodal approach, including pharmacology, neurostimulation and physiotherapy, could be beneficial in the management of patients with chronic CH. 

## Figures and Tables

**Figure 1 brainsci-09-00060-f001:**
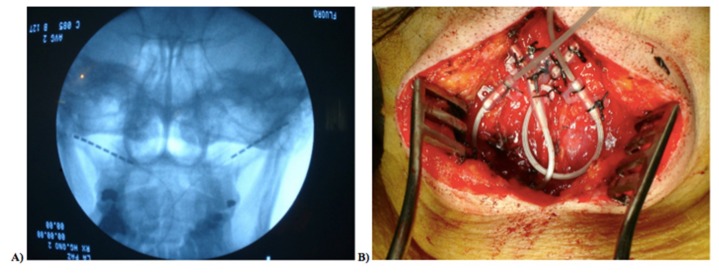
(**A**) Anteroposterior X-ray electrode image, after removing the needles, prepared for stimulation; (**B**) fascia anchor in suboccipital level.

**Figure 2 brainsci-09-00060-f002:**
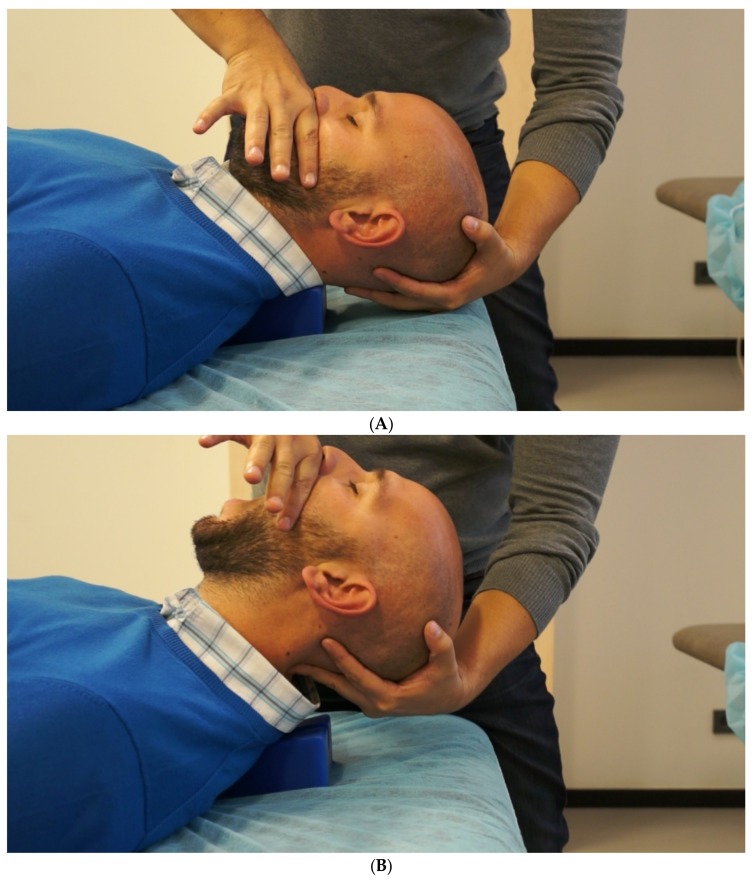
(**A**) Neural mobilization (step 1). Craniocervical flexion; (**B**) neural mobilization (step 2). Craniocervical extension plus mouth opening.

**Table 1 brainsci-09-00060-t001:** Data and % of change of physical variables and psychological characteristics.

	Pre	Post	3 months	4 months	% of change
**PPT**					**% Pre-4 months**
**V1 right**	1.12	0.75	1.71	1.49	33.03%
**V2 right**	1.62	1.22	2.45	2.77	70.98%
**V3 right**	1.39	0.94	2.5	1.67	20.14%
**Temporalis M1 right**	2.58	2.13	4.17	2.73	5.81%
**Temporalis M2 right**	3.28	3.19	5.7	5.16	57.32%
**V1 left**	0.72	0.62	1.52	1.29	79.17%
**V2 left**	1.27	1.08	2.86	3.1	144.09%
**V3 left**	1.37	0.88	3.03	2.41	75.91%
**Temporalis M1 left**	1.69	2.09	3.28	3.56	110.65%
**Temporalis M2 left**	2.85	3.19	5.35	4.64	62.81%
**Mastoid P right**	3.54	3.34	4.54	4.64	31.07%
**Mastoid P left**	2.81	2.14	3.52	4.2	49.47%
**Greater occipital N right**	4.75	3.8	4.68	4.04	−14.95%
**Greater occipital N left**	4.34	3.26	4.64	3.42	−21.20%
**Tibialis M right**	4.98	6.31	17.6	10.96	120.08%
**Tibialis M left**	5.84	5.17	17.08	12.68	117.12%
**Craniocervical flexion test**					
**time (s)**	3.06	8.22	12.47	24.06	-
**Fatigue**	21.5	22	43	60.5	-
**Physiological characteristic**					
**differences Pre-4 months**					
**HIT-6**	63	61	50	54	−9
**NDI**	18	12	13	13	−5
**PCS**	17	10	10	10	−7
**PCS rumination**	8	7	7	7	−1
**PCS magnification**	0	0	0	0	0
**PCS helplessness**	9	3	3	3	−6
**HDRS**	20	18	14	14	−6

PPT = pressure pain threshold; M = muscle; N = nerve; P = process; HIT-6 = headache impact test; NDI = neck disability index; PCS = pain catastrophizing scale; HDRS = Hamilton depression rating scale.

**Table 2 brainsci-09-00060-t002:** Evolution of headache diary.

Months	Frequency	Intensity	Duration	Abortive Treatment
May	2	9	25	SS-OT
June	3	9.33	23.33	SS
July	7	9.71	31.43	SS-OT
August	2	9.50	35	SS
September	0	-	-	-
October	0	-	-	-
November	4	8.50	33.75	SS-OT

SS = subcutaneous sumaptriptan; OT = oxygen therapy.

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
