# Peer review of "Endogenous Neurostimulation and Physiotherapy in Cluster Headache: A Clinical Case"

_brainsci, 2019, doi:10.3390/brainsci9030060_

Round 1
Reviewer 1 Report
Originality: This is a very short paper, providing a minimum of information about a case study, and constitutes a very minor contribution to the literature. The introduction section did not provide a clear rationale for carrying out the study (for example, why is your research question important? What gap in the literature is the study addressing?). I suggest to describe in this section only with the information related with the state of art related with endogenous neurostimulation and physiotherapy in cluster headache.
Methodologically Sound: As a case study report it is rather hard to go wrong methodologically, and the paper conforms to the standard.
Follows Appropriate Ethical Guidelines: Whilst there is no obvious declaration of ethical approval, it would appear to be a report of actions taken as part of normal clinical practice (as a case study report), and thus is acceptable.
Has results which are clearly presented and support the conclusions: Again, it conforms to the usual format for the presentation of a case study, although the content is very sparce. It is, however, appropriate enough, and does report a rare case likely to be of interest to a healthcare audience.
Overall Scientific Quality: As a minor case study report it lacks scientific depth, but effectovely is intended only to report the occurence of a typical case and to highlight the importance of correct disgnosis, and on these grounds merits attention.
Presentation, Organization, Clarity: I think you have some good information. But it is poorly presented. You will have to totally rewrite the manuscript. Make sure that you pay attention to language issues. Whilst I appreciate that the authors first language may not be English, there are many grammatical inaccuracies throughout the manuscript. It makes it very difficult to read and the text does not make sense in places. It is not the role of the reviewer to correct grammatical errors and it does disrupt the reading of the manuscript, which means the reviewers are less likely to give it a favourable review. I would suggest that you liaise with a colleague who is used to publishing in English to check any manuscript prior to submission as for example Tom Lang Communications and Training.
Correctly References Previous Relevant Work: It appears to reference prior work succinctly and accurately.
Importance/Interest: Although marked by its brevity, the content is of interest, particularly to clinicians such as neurologist who examine headache a great deal of the time, and physiotherpists who may need to be aware of the variant forms of this illnes.
OVERALL ASSESSMENT: (Please check statement that most clearly applies)
Some important information; needs major change; low priority
Author Response
REVIEWER 1
Originality: This is a very short paper, providing a minimum of information about a case study, and constitutes a very minor contribution to the literature.
The introduction section did not provide a clear rationale for carrying out the study (for example, why is your research question important? What gap in the literature is the study addressing?). I suggest to describe in this section only with the information related with the state of art related with endogenous neurostimulation and physiotherapy in cluster headache.
RESPONSE: We agree that there is several information in the literature about neurostimulation and cluster headache and about hyperalgesia and cluster headaches in different descriptive studies.
However, when we use this search strategy ("cluster headache"[MeSH Terms] AND physiotherapy[Title/Abstract]) in data bases, we only found 1 article and this article is related to Tension Type Headache (Torelli P, Jensen R, Olesen J. Physiotherapy for tension-type headache: a controlled study. Cephalalgia. 2004 Jan;24(1):29-36). That is why present information about “endogenous neurostimulation and physiotherapy in cluster headache” as you say is very, very poor nowadays. And again, that is why we present a single case that begins to cover the gap in the literature.
Clinically, we observed improvements in cluster headache patients when they are treated combining neurostimulation and physiotherapy technics. And this paper tries to open new frontiers in this field.
Methodologically Sound: As a case study report it is rather hard to go wrong methodologically, and the paper conforms to the standard.
RESPONSE: Thank you!
Follows Appropriate Ethical Guidelines: Whilst there is no obvious declaration of ethical approval, it would appear to be a report of actions taken as part of normal clinical practice (as a case study report), and thus is acceptable.
RESPONSE: Thank you!
Has results which are clearly presented and support the conclusions: Again, it conforms to the usual format for the presentation of a case study, although the content is very sparce. It is, however, appropriate enough, and does report a rare case likely to be of interest to a healthcare audience.
RESPONSE: Thank you again!
Overall Scientific Quality: As a minor case study report it lacks scientific depth, but effectovely is intended only to report the occurence of a typical case and to highlight the importance of correct disgnosis, and on these grounds merits attention.
RESPONSE: We agree.
Presentation, Organization, Clarity: I think you have some good information. But it is poorly presented. You will have to totally rewrite the manuscript. Make sure that you pay attention to language issues. Whilst I appreciate that the authors first language may not be English, there are many grammatical inaccuracies throughout the manuscript. It makes it very difficult to read and the text does not make sense in places. It is not the role of the reviewer to correct grammatical errors and it does disrupt the reading of the manuscript, which means the reviewers are less likely to give it a favourable review. I would suggest that you liaise with a colleague who is used to publishing in English to check any manuscript prior to submission as for example Tom Lang Communications and Training.
RESPONSE: Thank you for your comment and your offer. However for this paper, we used a professional writing and translation service for Health Professionals. If you need the invoice, we can show you.
Correctly References Previous Relevant Work: It appears to reference prior work succinctly and accurately.
RESPONSE: Thank you!
Importance/Interest: Although marked by its brevity, the content is of interest, particularly to clinicians such as neurologist who examine headache a great deal of the time, and physiotherpists who may need to be aware of the variant forms of this illnes.
RESPONSE: We agree!
Reviewer 2 Report
The article is well written. I have to congratulate the authors’ great work. There are several minor comments:
1. In the introduction, there are too many paragraphs addressing the effectiveness of oral medication for cluster headache. I suggest shortening them as those are not the main points of this article.
2. In the introduction, the authors mentioned SPG. If I am not wrong, is SPG mainly used for trigeminal neuralgia? Please clarify it.
3. Line 103: what is CARE checklist? Please clarify it.
4. Line 114: Since neuro-stimulation is the main point of this draft, I suggest the authors detailing how the greater occipital nerve complex is identified. Regarding the anatomy of the greater occipital nerve, please cite the following reference: Sonographic Nerve Tracking in the Cervical Region: A Pictorial Essay and Video Demonstration. AJPMR 2016.
5. Nowadays, ultrasound has been widely used in assessing neuromuscular disorders. For headache, it could result from chronic tension of the para-spinal and sub-occipital muscles, which can be easily checked by using ultrasound. Unfortunately, the authors did not use it to examine possible musculoskeletal painful origins. Please simply acknowledge it as a limitation by citing the following references: 1). Limb muscle quality and quantity in elderly adults with dynapenia but not sarcopenia: An ultrasound imaging study. Exp Gerontol. 2018
2). Static and Dynamic Shoulder Imaging to Predict Initial Effectiveness and Recurrence After Ultrasound-Guided Subacromial Corticosteroid Injections. Arch Phys Med Rehabil. 2017. 3). Basis of Shoulder Nerve Entrapment Syndrome: An Ultrasonographic Study Exploring Factors Influencing Cross-Sectional Area of the Suprascapular Nerve. Front Neurol. 2018
Author Response
REVIEWER 2
The article is well written. I have to congratulate the authors’ great work. There are several minor comments:
1. In the introduction, there are too many paragraphs addressing the effectiveness of oral medication for cluster headache. I suggest shortening them as those are not the main points of this article.
RESPONSE: Thank you for your comments. We have reduced some phrases about it and we have adapted the text according your comment.
2. In the introduction, the authors mentioned SPG. If I am not wrong, is SPG mainly used for trigeminal neuralgia? Please clarify it.
RESPONSE: There is several literatures about Sphenopalatine ganglion stimulation in patients with cluster headache. This is one of the most representative article about it: Jürgens, T.P.; Barloese, M.; May, A.; Láinez, J.M.; Schoenen, J.; Gaul, C.; Goodman, A.M.; Caparso, A.; Jensen, R.H. Long-term effectiveness of sphenopalatine ganglion stimulation for cluster headache. Cephalalgia 2017, 37, 423–434.
3. Line 103: what is CARE checklist? Please clarify it.
RESPONSE: It has been clarified and referenced in the manuscript.
4. Line 114: Since neuro-stimulation is the main point of this draft, I suggest the authors detailing how the greater occipital nerve complex is identified. Regarding the anatomy of the greater occipital nerve, please cite the following reference: Sonographic Nerve Tracking in the Cervical Region: A Pictorial Essay and Video Demonstration. AJPMR 2016.
RESPONSE: We have introduced this suggest.
5. Nowadays, ultrasound has been widely used in assessing neuromuscular disorders. For headache, it could result from chronic tension of the para-spinal and sub-occipital muscles, which can be easily checked by using ultrasound. Unfortunately, the authors did not use it to examine possible musculoskeletal painful origins. Please simply acknowledge it as a limitation by citing the following references:
1). Limb muscle quality and quantity in elderly adults with dynapenia but not sarcopenia: An ultrasound imaging study. Exp Gerontol. 2018
2). Static and Dynamic Shoulder Imaging to Predict Initial Effectiveness and Recurrence After Ultrasound-Guided Subacromial Corticosteroid Injections. Arch Phys Med Rehabil. 2017
3). Basis of Shoulder Nerve Entrapment Syndrome: An Ultrasonographic Study Exploring Factors Influencing Cross-Sectional Area of the Suprascapular Nerve. Front Neurol. 2018
RESPONSE: We have adapted this comment to the manuscript.
Round 2
Reviewer 1 Report
The paper has much improved, and although I have reservations about the interpretation of the data, and the strength of evidence for the clinical message, I think the article presents the data well enough for readers to judge themselves. I would recommend publication.